# NSAIDs, Ileal Inflammation, and Glucose Metabolism: Insights from a Large Retrospective Cohort

**DOI:** 10.3390/nu17091514

**Published:** 2025-04-29

**Authors:** Stephanie Hosanna Rodriguez, Gilles Jadd Hoilat, Nikash Pradhan, Carolina Gonzalez Bravo, Marcelo L. G. Correia, Mohamad Mokadem

**Affiliations:** 1Department of Internal Medicine, University of Minnesota, Minneapolis, MN 55455, USAgilles-hoilat@uiowa.edu (G.J.H.); marcelo-correia@uiowa.edu (M.L.G.C.); 2Division of Gastroenterology and Hepatology, University of Iowa Hospitals & Clinics, Iowa City, IA 52242, USA; 3Department of Internal Medicine, University of Iowa Carver College of Medicine, Iowa City, IA 52246, USA; 4Division of Endocrinology and Metabolism, University of Iowa Hospitals & Clinics, Iowa City, IA 52242, USA; 5Iowa City VA Health Care System, Iowa City, IA 52242, USA

**Keywords:** ileitis, inflammatory bowel disease (IBD), non-steroidal anti-inflammatory drugs (NSAIDs), type 2 diabetes mellitus (T2DM)

## Abstract

Background/Objectives: Ileitis, or inflammation of the terminal ileum, is often linked to inflammatory bowel disease (IBD), especially Crohn’s disease, but may also arise from non-steroidal anti-inflammatory drug (NSAID) use. While NSAIDs are known to cause gastrointestinal injury, their role in ileitis and downstream metabolic consequences remains unclear. This study evaluated the relationship between NSAID use, biopsy-confirmed ileitis, and glucose metabolism abnormalities in patients with and without IBD. Methods: We conducted a retrospective cohort study of 3725 adults who underwent ileal biopsy between 2009 and 2022 at a tertiary care center. Patients were stratified based on histologic evidence of ileitis. Collected data included demographics, IBD status, NSAID and steroid use, hemoglobin A1C, fasting glucose, and diagnoses of abnormal glucose metabolism. Multivariable logistic and linear regression models adjusted for age, BMI, sex, steroid use, and IBD. Results: Of 3725 patients, 876 had biopsy-confirmed ileitis. NSAID use—categorized as current, historical, or inpatient—was not significantly associated with ileitis after adjustment. In contrast, IBD was the strongest independent predictor (*p* < 0.05). Although unadjusted analyses showed lower A1C in the ileitis group (*p* = 0.003), this was not significant after controlling for confounders (*p* = 0.084). No significant associations were found between ileitis and fasting glucose or abnormal glucose metabolism. Age and BMI were the dominant predictors of glycemic abnormalities. Conclusions: NSAID use was not associated with biopsy-confirmed ileitis or impaired glucose metabolism. Traditional metabolic risk factors were stronger predictors of glycemic abnormalities than localized ileal inflammation.

## 1. Introduction

Ileitis, or inflammation of the ileum, is commonly associated with Crohn’s disease, a subtype of inflammatory bowel disease (IBD). However, it can also arise from infections, ischemia, neoplasms, vasculitides, and medication use, particularly non-steroidal anti-inflammatory drugs (NSAIDs) [1]. Given its diverse causes, timely and accurate diagnosis is crucial to avoid delays and mismanagement [2]. While ileitis is primarily recognized for its gastrointestinal (GI) effects, emerging evidence suggests it may also contribute to systemic health complications, including metabolic dysregulation in both IBD and non-IBD populations.

NSAIDs are among the most widely used medications for managing pain, inflammation, arthritis, and fevers, as well as for colorectal cancer prevention [3]. They work by inhibiting cyclooxygenase (COX-1 and COX-2) enzymes, which regulate prostaglandin synthesis—key mediators of inflammation and pain [4,5,6]. Despite their widespread use, NSAIDs are well known for their GI complications, including NSAID-induced enteropathy, which can contribute to or exacerbate intestinal inflammation, such as ileitis.

While upper GI complications, such as gastric and duodenal ulcers, are well documented, NSAIDs have also been linked to lower GI issues, including colonic bleeding, strictures, protein loss, and anemia [7]. In patients with IBD, NSAIDs may aggravate preexisting inflammation, whereas in non-IBD patients, they can induce de novo inflammation, including ileitis [1,4]. Pathology-based comparisons further emphasize that NSAID-induced ileitis is morphologically distinct from Crohn’s disease. NSAID enteropathy is often characterized by crypt atrophy, diminished Paneth cells, and reduced lamina propria inflammation—features not typically seen in IBD [8]. Di Lauro and Crum-Cianflone proposed that the shift toward extended-release and enteric-coated NSAID formulations may have altered the pattern of GI injury. By bypassing the stomach and dissolving further along the digestive tract, these formulations increase the risk of damage in the distal small intestine and colon, with pill retention at the ileocecal valve potentially contributing to ileitis [1]. Even enteric-coated aspirin, designed to limit gastric injury, has been shown to cause mucosal breaks and red spots in the ileum, further challenging the assumption that delayed-release formulations provide meaningful protection to the distal gut [9].

Although NSAID-induced inflammation is relatively uncommon, affected patients may present with erosions, ulcerations, bleeding, perforation, strictures, or obstructions [1,10]. Diagnosis typically involves direct visualization through capsule endoscopy or enteroscopy, with key criteria including a history of NSAID use, absence of recent antimicrobial therapy, and negative stool or tissue cultures for bacterial infections [3,11,12]. Research on NSAID-induced ileitis has produced inconsistent findings, potentially due to variability in diagnostic methods. While some studies rely on imaging or endoscopic visualization without tissue sampling, our study uses biopsy-confirmed ileitis to provide a more definitive assessment. This histological approach minimizes misclassification and addresses diagnostic ambiguity, allowing us to more accurately evaluate whether NSAID use is truly associated with ileal inflammation.

In addition to clarifying this relationship, our study also explores potential systemic consequences of ileitis—particularly its role in metabolic dysregulation. While some NSAID-related metabolic effects have been hypothesized to occur through alternative pathways, such as modulation of intestinal taste and glucose-sensing receptors, our focus centers on mechanisms specifically linked to ileal inflammation [13]. Preclinical work by Herz et al. showed that NSAID-induced ileal injury impairs GLP-1 secretion and disrupts glucose homeostasis. Given GLP-1’s central role in insulin regulation, we investigate whether biopsy-confirmed ileitis correlates with metabolic markers such as elevated hemoglobin A1C, fasting glucose, and diagnoses of prediabetes or diabetes in both IBD and non-IBD populations. By integrating histological confirmation with clinical data, we aim to clarify both the etiologic role of NSAIDs and the downstream metabolic implications of ileal inflammation.

## 2. Materials and Methods

### 2.1. Study Design and Population

This retrospective cohort study utilized electronic health records (EHR) from the University of Iowa Hospitals and Clinics, with data spanning from 1 January 2009 to 31 December 2022. Patient data were retrieved through the Institute for Clinical and Translational Science (ICTS) using the Epic medical records system. The study included adult patients (≥18 years), both living and deceased, who underwent ileal biopsy during the specified period. Patients were stratified into two cohorts: those with biopsy-confirmed ileitis and those without ileitis (control group).

### 2.2. Inclusion and Exclusion Criteria 

Adult patients aged 18 years and older with available ileal biopsy results were included in this study. Both living and deceased patients were eligible for analysis to ensure a comprehensive evaluation. Additionally, patients diagnosed with IBD, including Crohn’s disease and ulcerative colitis (UC), were incorporated into the analysis. Exclusion criteria included pediatric patients under 18 years of age and individuals without documented ileal biopsy results. These criteria were established to maintain the study’s focus on adult populations with confirmed histological data.

### 2.3. Data Collection and Variables

Patient demographic and clinical data were collected from electronic health records (EHR), including age, gender, body mass index (BMI), and medication use. Demographic data were categorized by sex, age (18–30, 31–50, 51–70, 71+ years), and BMI (underweight: <18.5, normal weight: 18.5–24.9, overweight: 25–29.9, obese: ≥30). Clinical outcomes assessed included the presence of ileitis (confirmed by ileum biopsy), IBD status (diagnosed as Crohn’s disease or UC), and NSAID use.

NSAID use was categorized into three types: current NSAIDs (NSAIDs prescribed for more than 5 days within 6 months prior to the biopsy, not necessarily indicating active use), historical NSAIDs (patient-reported NSAID use within 6 months prior to biopsy), and inpatient NSAIDs (NSAID administration for more than 5 days during inpatient care within 6 months before the biopsy). NSAID medications included a range of oral-only drugs such as aspirin, ibuprofen, naproxen, and others, along with historical use of rofecoxib and valdecoxib (both of which have been withdrawn from the market). Steroid use, both oral and intravenous, within 3 months prior to the biopsy was also recorded, with commonly prescribed steroids including prednisone, methylprednisolone, budesonide, and hydrocortisone.

Laboratory data included A1C levels (measured within 3 months before or after biopsy), fasting glucose levels (measured within 6 months before or after biopsy), and diagnoses of abnormal glucose metabolism (e.g., prediabetes, diabetes). Clinical outcomes included A1C levels, which reflect long-term blood glucose control, fasting glucose levels, and abnormal glucose metabolism as determined by diagnoses of prediabetes (R73.03), diabetes (E11.9), or other glucose abnormalities (R73.09).

### 2.4. Statistical Analysis

All statistical analyses were performed using SPSS Statistics (version 28.0). The primary aim of the analysis was to explore the relationships between NSAID use, ileitis, and metabolic outcomes (A1C levels, fasting glucose levels, and abnormal glucose metabolism), adjusting for potential confounders such as BMI, age, gender, and steroid use.

Descriptive statistics were used to summarize the demographic and clinical characteristics of the ileitis and control groups. Continuous variables (e.g., age, BMI, A1C, fasting glucose) were presented as means with standard deviations, while categorical variables (e.g., gender, NSAID use) were reported as frequencies and percentages. Independent t-tests were applied to compare continuous variables (e.g., A1C levels, fasting glucose) between the ileitis and control groups, stratified by IBD status. Chi-square tests examined associations between categorical variables (e.g., NSAID use, abnormal glucose metabolism) and the presence of ileitis.

Multivariable logistic regression was used to evaluate the relationships between NSAID use (current, historical, inpatient) and the presence of ileitis, as well as metabolic outcomes (A1C, fasting glucose, abnormal glucose metabolism), adjusting for factors such as IBD diagnosis, BMI, age, gender, and steroid use. Additionally, multivariable linear regression was conducted to investigate the association between ileitis and A1C levels and fasting glucose, excluding steroid use and controlling for BMI, age, and gender. These regression analyses aimed to control for potential confounders and assess the impact of ileitis on glucose-related outcomes.

Interaction effects between NSAID use and BMI, age, and gender were explored to determine whether these factors modified the relationship between NSAID use and ileitis. Statistical significance was set at *p* < 0.05, with 95% confidence intervals (CIs) calculated for odds ratios (ORs) in logistic regression models and coefficients in linear regression models.

Outliers were removed based on pre-defined thresholds to ensure data quality and minimize the influence of extreme values. This approach aligns with established practices in metabolic and cardiovascular research, where such thresholds are used to identify and remove biologically implausible or non-representative data points [14,15]. BMI values exceeding three standard deviations from the mean were excluded, resulting in the removal of 30 values from the ileitis group and 17 from the control group. Similarly, A1C values beyond three standard deviations led to the exclusion of one value from the ileitis group and two from the control group. Additionally, fasting glucose values below 40 mg/dL were removed, affecting four participants in the ileitis group and nine in the control group.

## 3. Results

### 3.1. Demographics and Clinical Characteristics

A total of 3725 participants were included in the study: 876 in the ileitis group and 2849 in the control group. The mean age was slightly higher in the ileitis group (42.87 years, SD = 16.12) than in the control group (41.67 years, SD = 16.17), though this difference was not statistically significant (*p* = 0.055). BMI was marginally lower in the ileitis group (mean = 27.78, SD = 8.10) compared to the control group (mean = 28.42, SD = 8.73), with a *p*-value of 0.054. An IBD diagnosis code (Crohn’s disease or UC) was present in 93.4% of patients in the ileitis group and 78.3% of controls (*p* < 0.001). Diagnosis codes were unavailable or absent in approximately 7% of ileitis cases and 21.7% of controls. Conversely, abnormal glucose metabolism was more frequent in the control group (33.2%) than in the ileitis group (21.3%, *p* < 0.001), Table 1. 

### 3.2. NSAID Use and Ileitis

NSAID use—whether current, historical, or inpatient—was not significantly associated with the presence of ileitis. In the ileitis group, 72.5% were current NSAID users, 25.0% reported historical NSAID use, and 5.0% had inpatient NSAID use. These rates were comparable to the control group (69.4%, 29.1%, and 6.3%, respectively). None of the comparisons reached statistical significance (*p* > 0.05). Logistic regression models, adjusted for IBD, steroid use, sex, age, and BMI, confirmed that IBD was the only significant predictor of ileitis in all models. NSAID use (in any form) did not predict the presence of ileitis, Table 2.

### 3.3. A1C Levels in Ileitis Versus Control Group

A1C levels were significantly lower in the ileitis group compared to the control group. Specifically, the mean A1C in the ileitis group was 5.75% (SD = 0.91), while in the control group it was 6.22% (SD = 1.55). This difference was statistically significant (*p* = 0.003), with a moderate effect size (Cohen’s d = 0.52). However, when controlling for abnormal glucose metabolism, age, BMI, sex, and IBD in a multiple regression model, ileitis was no longer a significant predictor of A1C (*p* = 0.084). Abnormal glucose metabolism was the only significant predictor (*p* < 0.001), showing strong positive association with elevated A1C, Table 3 and Table 4.

### 3.4. Fasting Glucose Levels in Ileitis Versus Control Group

Fasting glucose levels were slightly lower in the ileitis group (mean = 99.64 mg/dL) compared to the control group (mean = 108.87 mg/dL), but this difference was not statistically significant (*p* = 0.095). When controlling for age, BMI, and sex in regression models, ileitis was not a significant predictor of fasting glucose levels, Table 5.

### 3.5. Abnormal Glucose Metabolism (Pre-Diabetes/Diabetes) in Ileitis Versus Control Group

While abnormal glucose metabolism was more prevalent in the control group, logistic regression, adjusting for confounders, showed that only age and BMI were independent predictors, Table 6.

### 3.6. A1C and Fasting Glucose in IBD with Ileitis

In participants with IBD, neither A1C nor fasting glucose levels were significantly influenced by the presence of ileitis. Regression models adjusting for age, sex, and BMI showed no significant association between ileitis and A1C levels (*p* = 0.147), and similar findings were observed for fasting glucose (*p* = 0.323), Table 7.

### 3.7. Abnormal Glucose Metabolism (Pre-Diabetes/Diabetes) in IBD with Ileitis

When restricted to participants with IBD, there was no significant association between ileitis and abnormal glucose metabolism. Logistic regression showed that age (OR = 1.052) and BMI (OR = 1.067) remained significant predictors, while ileitis, sex, and steroid use were not statistically significant, Table 8.

## 4. Discussion

This study evaluated the relationship between NSAID use, biopsy-confirmed ileitis, and systemic metabolic dysregulation in a large tertiary care cohort, adjusting comprehensively for BMI, age, gender, steroid use, and IBD status. Contrary to common clinical assumptions, NSAID use was not significantly associated with ileitis. Previous studies suggesting such a link have often relied on non-specific markers—such as symptoms, endoscopic appearance, or imaging—without histologic validation, potentially leading to overestimation of NSAID-related ileal injury [1,11,16,17]. By focusing on biopsy-confirmed inflammation, our study mitigates diagnostic misclassification and provides more definitive evidence against a causal role for NSAIDs in chronic ileitis [17].

While NSAIDs are known to transiently increase intestinal permeability and may induce acute mucosal injury, particularly in the small bowel [3,18], our findings suggest these changes are insufficient to result in persistent, histologically apparent inflammation unless compounded by other risk factors. In contrast, IBD emerged as the strongest predictor of ileitis, underscoring the importance of comprehensive diagnostic evaluation in patients presenting with ileal inflammation [8]. This finding reinforces the need to prioritize IBD as the primary differential in such cases, with careful exclusion of alternative etiologies such as medication-related injury [19].

We also explored the hypothesis that ileal inflammation could adversely impact glucose metabolism. Although patients with ileitis initially appeared to have lower A1C levels, this association disappeared after adjusting for confounders, particularly age and BMI—traditional determinants of metabolic health. Similar trends were seen for fasting glucose and prevalence of abnormal glucose metabolism. These findings suggest that localized ileal inflammation, independent of systemic factors, does not substantially impair glucose regulation.

Interestingly, preclinical studies have demonstrated that NSAID-induced ileal injury can impair GLP-1 secretion and disrupt glucose homeostasis [20,21]. The absence of a similar pattern in our clinical population may reflect differences in exposure duration, inflammatory severity, and host response. Unlike tightly controlled animal models, human populations introduce a wide array of metabolic and immunologic variability that may buffer or obscure localized effects of intestinal inflammation on systemic metabolism.

This study has important limitations that warrant mention. First, as a retrospective observational analysis, it is inherently subject to potential residual confounding despite multivariable adjustment for key covariates. Second, NSAID exposure was determined through electronic health records, which may not fully capture over-the-counter use, exact dosing, duration, or formulation—factors that could influence GI outcomes. Third, due to the nature of the dataset, we were unable to evaluate mechanistic pathways, including GLP-1 secretion, microbiome alterations, or other biomarkers that may link ileal inflammation to glucose metabolism. Lastly, the study was conducted at a single tertiary care center, which may limit the generalizability of findings to broader or more heterogeneous populations.

Looking ahead, future research should pursue prospective, longitudinal studies to elucidate causal relationships between GI inflammation, NSAID exposure, and systemic metabolic dysfunction. Such studies would benefit from direct assessments of gut-derived metabolic regulators (e.g., GLP-1), insulin sensitivity metrics, microbiome composition, and inflammatory cytokine profiles, all of which could uncover subtle or subclinical mechanistic pathways. Moreover, patient stratification by IBD subtype, disease duration, histological severity, and treatment history will be essential for advancing personalized clinical strategies that target both intestinal inflammation and systemic metabolic health in an integrated manner.

## 5. Conclusions

Our study provides compelling evidence challenging the presumed causal role of NSAIDs in ileitis. Biopsy-confirmed ileitis did not independently correlate with impaired glucose metabolism after controlling for traditional metabolic risk factors. These findings support prioritizing the identification and management of underlying IBD and conventional metabolic risk factors when evaluating patients with ileitis.

## Figures and Tables

**Table 1 nutrients-17-01514-t001:** Demographics and clinical characteristics of the study population.

Characteristic	Control Group	Ileitis Group
Total Participants	2849	876
Female (n, %)	1714 (60.2%)	477 (54.5%)
Male (n, %)	1135 (39.8%)	399 (45.5%)
IBD Diagnosis Code (n, %)	1160 (78.3%)	642 (93.4%)
Abnormal Glucose Metabolism (n, %)	492 (33.2%)	146 (21.3%)
Current NSAID Use (n, %)	143 (69.4%)	29 (72.5%)
Historical NSAID Use (n, %)	60 (29.1%)	10 (25.0%)
Age (Mean ± SD)	41.67 ± 16.17	42.87 ± 16.12
BMI (Mean ± SD)	28.42 ± 8.73	27.78 ± 8.10
A1C (Mean ± SD)	6.23 ± 1.55	5.79 ± 0.90
A1C Range	4.30–12.00%	4.20–8.60%
Fasting Glucose (Mean ± SD)	108.87 ± 34.06 mg/dL	99.64 ± 13.32 mg/dL

Note. Values are presented as means ± standard deviations or n (%).

**Table 2 nutrients-17-01514-t002:** NSAID use and odds of ileitis.

NSAID Type	Ileitis Group	Control Group	*p*-Value	OR (Adjusted)	*p*-Value (Adjusted)
Current NSAID Use	72.5%	69.4%	0.697	1.96	0.294
Historical NSAID Use	25.0%	29.1%	0.597	0.61	0.437
Inpatient NSAID Use	5.0%	6.3%	0.751	0.57	0.646
IBD Diagnosis	93.4%	78.3%	<0.001 *	~7.00	0.011–0.039 *

Note: * *p* < 0.05.

**Table 3 nutrients-17-01514-t003:** Comparison of mean A1C levels between ileitis and control groups.

Variable	Ileitis Group	Control Group	*p*-Value
Mean A1C %	5.75 (SD = 0.91)	6.22 (SD = 1.55)	0.003

**Table 4 nutrients-17-01514-t004:** Multiple regression predicting A1C levels.

Predictor	B	SE	Beta	*p*-Value	95% CI for B
Abnormal Glucose Metabolism	1.10	0.303	0.291	<0.001 *	[0.502, 1.699]
Ileitis	−0.428	0.247	−0.120	0.084	[−0.914, 0.058]
IBD	−0.190	0.241	−0.063	0.431	[−0.666, 0.286]
Sex (Male)	0.204	0.199	0.067	0.305	[−0.188, 0.597]
Age	−0.002	0.007	−0.017	0.801	[−0.015, 0.012]
BMI	0.019	0.012	0.106	0.111	[−0.004, 0.042]

Note: B = unstandardized regression coefficient; SE = standard error; Beta = standardized regression coefficient; CI = confidence interval. Statistical significance is defined as * *p* < 0.05.

**Table 5 nutrients-17-01514-t005:** Comparison of mean fasting glucose levels between ileitis and control groups.

Variable	Ileitis Group	Control Group	*p*-Value
Mean Fasting Glucose (mg/dL)	99.64 (SD = 13.32)	108.87 (SD = 34.06)	0.095

**Table 6 nutrients-17-01514-t006:** Logistic regression predicting abnormal glucose metabolism.

Predictor	B	SE	OR	95% CI	*p*-Value
Ileitis	0.028	0.161	1.029	[0.750, 1.411]	0.861
Age	0.049	0.005	1.050	[1.040, 1.061]	<0.001 *
BMI	0.073	0.011	1.076	[1.053, 1.099]	<0.001 *
Steroid Use	−0.360	0.205	0.698	[0.467, 1.042]	0.079
Sex	−0.047	0.157	0.955	[0.702, 1.298]	0.766

Note: B = regression coefficient; SE = standard error; OR = odds ratio; CI = confidence interval. Statistical significance is defined as * *p* < 0.05.

**Table 7 nutrients-17-01514-t007:** Regression results for A1C in IBD subgroup.

Predictor	B	SE	Beta	*p*-Value	95% CI for B
Ileitis	−0.401	0.348	−0.136	0.253	[−1.095, 0.293]
Sex	0.132	0.335	0.046	0.696	[−0.537, 0.800]
Age	0.023	0.011	0.234	0.045 *	[0.001, 0.045]
BMI	0.025	0.022	0.138	0.260	[−0.019, 0.068]

Note: B = unstandardized regression coefficient; SE = standard error; Beta = standardized regression coefficient; CI = confidence interval. Statistical significance defined as * *p* < 0.05.

**Table 8 nutrients-17-01514-t008:** Predictors of abnormal glucose metabolism in IBD subgroup.

Predictor	B	SE	OR	95% CI	*p*-Value
Ileitis	0.080	0.145	1.083	[0.814, 1.440]	0.584
Sex	−0.009	0.141	0.991	[0.752, 1.306]	0.949
Age	0.051	0.005	1.052	[1.043, 1.062]	<0.001 *
BMI	0.065	0.010	1.067	[1.047, 1.087]	<0.001 *

Note: B = regression coefficient; SE = standard error; OR = odds ratio; CI = confidence interval. Statistical significance is defined as * *p* < 0.05.

## Data Availability

Data are not publicly available due to ethical and privacy restrictions. Requests for data access may be considered upon reasonable request to the corresponding author and subject to approval by the Institutional Review Board of the University of Iowa Hospitals & Clinics.

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
