# Peer review of "NSAIDs, Ileal Inflammation, and Glucose Metabolism: Insights from a Large Retrospective Cohort"

_nutrients, 2025, doi:10.3390/nu17091514_

Round 1
Reviewer 1 Report
Comments and Suggestions for Authors
The article by Stephanie Rodriguez, Gilles Jadd Hoilat, Nikash Pradhan, Carolina Gonzalez Bravo, Marcelo L. Correia, and Mohamad Mokadem MD, entitled “NSAIDs, Ileal inflammation, and Glucose Metabolism: Insights from a Large Retrospective Cohort”, aims to study to evaluate the relationship between NSAID use, biopsy-confirmed ileitis, and glucose metabolism abnormalities in patients with and without IBD.
I recommend that the paper be accepted after minor revision:
- The author should check for typographical and grammar error the entire manuscript (space, page lines etc…)
- Authors should reduce the abstract to increase readability and fluency.
- What are the limitations of the study? The authors are invited to add the limitations of the study at the end of the conclusions.
- Provide details on the use of NSAIDs among participants: how long and how often did patients use NSAIDs?
- It could be interesting to examine the distinct subgroups of IBD for a comparison of ileitis and glucose metabolism.
- More recent references are recommended
Author Response
Here’s our point-by-point responses in bold to Reviewer #1 comments
Comment (1): The author should check for typographical and grammar error the entire manuscript (space, page lines etc…)
Answer (1) We have carefully reviewed the manuscript for typographical, grammatical, and formatting issues. While we did identify and correct a minor spacing inconsistency near one section break, we did not observe any pervasive grammatical or typographic errors. To further ensure the manuscript’s clarity and quality, an English major independently reviewed the full draft and similarly did not identify issues with grammar or structure. Additionally, we closely followed the journal’s manuscript template to ensure consistency in formatting, paragraph spacing, and layout. If there are specific areas that appear problematic, we would be grateful if you could kindly indicate the page or section so we can address them more directly in our revision.
Comment (2): Authors should reduce the abstract to increase readability and fluency.
Answer (2): We have revised the abstract to improve clarity and conciseness, reducing its length while retaining all key methodological and outcome details. The updated version adheres to journal guidelines and enhances overall readability and fluency.
Comment (3): What are the limitations of the study? The authors are invited to add the limitations of the study at the end of the conclusions.
Answer (3): We have added a dedicated limitations paragraph near the end of the Discussion section, just before the concluding remarks. This new paragraph outlines key limitations of our study, including its retrospective design, the potential for misclassification of NSAID exposure, the absence of mechanistic, and the single-center setting which may affect generalizability.
Comment (4): Provide details on the use of NSAIDs among participants: how long and how often did patients use NSAIDs?
Answer (4): We have clarified our NSAID exposure definitions in the Methods section (Section 2.3). NSAID use was categorized into three groups: (1) Current NSAID use was defined as NSAID prescriptions for more than 5 days within the 6 months prior to biopsy; (2) Historical NSAID use included patient-reported NSAID use within the 6 months prior to biopsy but without active prescriptions; and (3) Inpatient NSAID use referred to administration of NSAIDs for more than 5 days during a hospitalization in the 6 months preceding biopsy. Due to the retrospective nature of the study, precise details regarding over-the-counter use, dosing frequency, duration beyond the 6-month window, and specific formulations were not consistently available. We have also noted this limitation explicitly in the revised Discussion section.
Comment (5): It could be interesting to examine the distinct subgroups of IBD for a comparison of ileitis and glucose metabolism.
Answer (5): We appreciate the reviewer’s insightful suggestion to explore differences across distinct IBD subgroups in relation to ileitis and glucose metabolism. Unfortunately, our dataset did not include sufficient granularity to reliably stratify patients by IBD subtype or disease activity. However, we agree that this would be a valuable direction for future studies, particularly those with larger cohorts and more detailed phenotyping.
Comment (6): More recent references are recommended
Answer (6): We have supplemented key sections of the manuscript such as the introduction, methods, and discussion with additional citations.

Reviewer 2 Report
Comments and Suggestions for Authors
The work presented by Rodriguez et al. is a very well written manuscript and comprehensively presented. They clearly define the subject of analysis and meticulously address all aspects of the study. They also reach some interesting conclusions from the statistical analysis. There are however important issues in relation to the discussion section. There is not a single reference to other researchers work in the whole discussion section – see for example two situations reported in major comments (ii). Five paragraphs of discussion and zero bibliographic references. This section needs considerable work from the authors to backup their major conclusions in the study.
Major comments:
- Pages 3-4, lines 148-153: The authors need to justify their chosen pre-defined thresholds. Was this solely based on generic statistical assumptions or was there any biological reasoning for adopting such thresholds?
- Page 6 lines 218-221: The mention in this sentence - “Previous studies suggesting such a link …”, however they do not provide and reference to which studies they are referring to. Being a discussion section, one would expect that comparisons would be made to other existing studies and that those studies would be cited. Same for the paragraph that follows in discussion section - “While NSAIDs are known to transiently increase …”; not a single reference is used to support the comment made by the authors.
Author Response
Here’s our point-by-point responses in bold to Reviewer #2 comments
Comment (1): Pages 3-4, lines 148-153: The authors need to justify their chosen pre-defined thresholds. Was this solely based on generic statistical assumptions or was there any biological reasoning for adopting such thresholds?
Answer (1): We have revised the Methods section (pages 3–4, lines 148–153) to clarify that values exceeding ±3 standard deviations from the mean for BMI, A1C, and glucose were excluded to reduce the influence of biologically implausible outliers. This is a standard data-cleaning approach in metabolic research. We have added supporting references where similar thresholds were applied.
Comment (2): Page 6 lines 218-221: The mention in this sentence - “Previous studies suggesting such a link …”, however they do not provide and reference to which studies they are referring to. Being a discussion section, one would expect that comparisons would be made to other existing studies and that those studies would be cited. Same for the paragraph that follows in discussion section - “While NSAIDs are known to transiently increase …”; not a single reference is used to support the comment made by the authors.
Answer (2): We have revised the discussion section to incorporate citations throughout, including references to prior studies examining NSAID-induced ileitis, intestinal permeability, GLP-1 secretion, and IBD-related inflammation. These references now support our interpretation of the findings and help contextualize our results within the broader literature. We hope these additions adequately address the reviewer’s co

Reviewer 3 Report
Comments and Suggestions for Authors
Rodriguez et al. evaluated the association of NSAID use and glucose metabolism with ileitis. Since this is a human study, biopsy samples were obtained from patients with symptoms of intestinal disease. Consequently, comparison of NSAID use and variables associated with glucose metabolism in patients with and without intestinal disease could not be made. Therefore, comparison of inflammation of the ileum and NSAID use and variables associated with glucose metabolism is reasonable. However, some minor changes in the manuscript need to be made before publication.
The Abstract concludes with "These findings underscore the importance of considering traditional metabolic risk factors over localized ileal inflammation when evaluating systemic metabolic dysfunction." While the study did assess NSAID use and glucose metabolism, the focus of the study was not systemic metabolic dysfunction, i.e., patients cohorts with and without systemic metabolic dysfunction were not studied. Therefore, the concluding sentence should more closely follow the Conclusions section of the manuscript. If the authors consider it necessary, a paragraph discussing systemic metabolic dysfunction can be added to the Discussion.
Section 3.1 states "IBD was more prevalent in the ileitis group (93.4%) compared to the control group (78.3%, p < 0.001)."
This may confuse some readers: In the Introduction, ileitis is defined as inflammation of the ileum, and all of the patients in the ileitis group had biopsy-confirmed ileitis. Therefore, 100% of the patients in the ileitis group had inflammation of the ileum, suggesting that 100% of the patients in the ileitis group had inflammatory bowel disease.
Also, the reason why 1160 patients in the control group (2849 patients) is 78.3% of the control group and 642 patients in the ileitis group (876 patients) is 93.4% of the ileitis group should be mentioned.
If correct, I suggest changing the sentence to something like
"Diagnosis of Crohn’s disease (CD) or ulcerative colitis (UC) was more prevalent in the ileitis group (93.4%) compared to the control group (78.3%, p < 0.001): diagnosis of the absence or presence of CD or UC in 1368 patients in the control group and 189 patients in the ileitis group was not available."
If this is correct, then Table 1 "IBD Diagnosis (n, %)" needs to be changed to something like "Diagnosis of CD/UC (n, %)".
If the above is not correct, then diagnosis of inflammation of the ileum that is not associated with inflammatory bowel disease needs to be explained.
In Table 3, the authors should consider changing Mean A1C (%) to Mean A1C %, because the values in parentheses in Table 3 are the SDs.
All abbreviations should be defined. In Tables 4, 5, 6, and 7 "B", "SE", and "Beta" need to be defined. This can be done in a footnote. OR: Odds Ratios, can also be added to these footnotes. If the authors prefer, the definitions of the abbreviations can be added to the Abbreviations section instead of the Table footnotes.
The Discussion states "We also explored the hypothesis that ileal inflammation could adversely impact glucose metabolism through pathways involving gut hormones like GLP-1 or microbiota-mediated metabolic shifts."
However, GLP-1 and microbiota-mediated metabolic shifts were not specifically investigated in the current study. Therefore, the phrase "through pathways involving gut hormones like GLP-1 or microbiota-mediated metabolic shifts" needs to be deleted: "We also explored the hypothesis that ileal inflammation could adversely impact glucose metabolism."
The Discussion states "Interestingly, preclinical studies have demonstrated that NSAID-induced ileal injury can impair GLP-1 secretion and disrupt glucose homeostasis. The absence of a similar pattern in our clinical population may reflect differences in exposure duration, inflammatory severity, and host response."
Again, GLP-1 was not specifically investigated in the current study. Therefore, this passage should be changed to something like
"Interestingly, preclinical studies have demonstrated that NSAID-induced ileal injury can impair GLP-1 secretion and disrupt glucose homeostasis. The absence of disrupted glucose homeostasis in our clinical population may reflect differences in exposure duration, inflammatory severity, and host response."
The conclusions state "Our study provides compelling evidence challenging the presumed causal role of NSAIDs in ileitis, highlighting instead the dominant etiologic importance of IBD."
The passage "highlighting instead the dominant etiologic importance of IBD" appears to be redundant since inflammatory bowel disease would appear to obviously be an important etiologic factor in inflammation of the ileum. Therefore, the sentence should be changed to something like
"Our study provides compelling evidence challenging the presumed causal role of NSAIDs in ileitis", which is the primary finding of the study.
Author Response
Here’s our point-by-point responses in bold to Reviewer #3 comments
Comment (1):
"The Abstract concludes with 'These findings underscore the importance of considering traditional metabolic risk factors over localized ileal inflammation when evaluating systemic metabolic dysfunction.' While the study did assess NSAID use and glucose metabolism, the focus of the study was not systemic metabolic dysfunction... Therefore, the concluding sentence should more closely follow the Conclusions section of the manuscript."
Answer (1): We appreciate this suggestion and have revised the Abstract conclusion to better reflect the scope and structure of the study. The revised sentence now reads:
“Traditional metabolic risk factors were stronger predictors of glycemic abnormalities than localized ileal inflammation.”
While our study did not enroll patients based on predefined syndromes such as metabolic syndrome or insulin resistance, we did evaluate the presence of abnormal glucose metabolism based on documented diagnoses of prediabetes (ICD-10 code R73.03), type 2 diabetes mellitus (E11.9), and other abnormal glucose states (R73.09). These diagnoses were supported by laboratory data including A1C and fasting glucose levels. We agree that “systemic metabolic dysfunction” could imply broader syndrome-level evaluation, so we have refined our language to clarify that we focused on clinically coded glycemic disorders and metabolic risk markers.
Comment (2):
"Section 3.1 states 'IBD was more prevalent in the ileitis group (93.4%)...' This may confuse some readers... If correct, I suggest changing the sentence... Also, Table 1 'IBD Diagnosis (n, %)' needs to be changed."
Answer (2): We clarified the text in Section 3.1 as follows:
“An IBD diagnosis code (Crohn’s disease or Ulcerative colitis) was present in 93.4% of patients in the ileitis group and 78.3% of controls (p < 0.001). Diagnosis codes were unavailable or absent in approximately 7% of ileitis cases and 21.7% of controls.”
We also updated Table 1 to reflect this clarification. The variable now reads:
“IBD Diagnosis Code (n, %).”
Comment (3):
"In Table 3, the authors should consider changing Mean A1C (%) to Mean A1C %, because the values in parentheses in Table 3 are the SDs."
Answer (3): Revised as suggested. Table 3 now labels the column as “Mean A1C %.”
Comment (4):
"All abbreviations should be defined. In Tables 4, 5, 6, and 7 'B', 'SE', and 'Beta' need to be defined..."
Answer (4):
We have added standardized footnotes to each relevant table defining the abbreviations used. For example:
- Tables 4 and 7
Note: B = unstandardized regression coefficient; SE = standard error; Beta = standardized regression coefficient; CI = confidence interval. Statistical significance defined as p < 0.05.
- Tables 6 and 8:
Note: B = regression coefficient; SE = standard error; OR = odds ratio; CI = confidence interval. Statistical significance defined as p < 0.05.
Comment (5):
"The Discussion states 'We also explored the hypothesis that ileal inflammation could adversely impact glucose metabolism through pathways involving gut hormones like GLP-1 or microbiota-mediated metabolic shifts.' However, GLP-1 and microbiota-mediated shifts were not specifically investigated..."
Answer (5): We agree and have revised the sentence to remove references to mechanisms not directly studied. It now reads:
“We also explored the hypothesis that ileal inflammation could adversely impact glucose metabolism.”
Comment (6):
"The conclusions state 'Our study provides compelling evidence challenging the presumed causal role of NSAIDs in ileitis, highlighting instead the dominant etiologic importance of IBD.' The phrase regarding IBD appears redundant..."
Answer (6):
We revised the sentence for clarity. This change removes redundancy while preserving the study’s key message. The new conclusion reads:
“Our study provides compelling evidence challenging the presumed causal role of NSAIDs in ileitis.”

Round 2
Reviewer 2 Report
Comments and Suggestions for Authors
The authors answered to the raised comments satisfactorily.